# Prognostic Value of Skeletal Muscle Loss in Patients with Hepatocellular Carcinoma Treated with Hepatic Arterial Infusion Chemotherapy

**DOI:** 10.3390/cancers15061834

**Published:** 2023-03-18

**Authors:** Kyoko Oura, Asahiro Morishita, Joji Tani, Takako Nomura, Takushi Manabe, Kei Takuma, Mai Nakahara, Tomoko Tadokoro, Koji Fujita, Shima Mimura, Takayuki Sanomura, Yoshihiro Nishiyama, Tsutomu Masaki

**Affiliations:** 1Department of Gastroenterology and Neurology, Faculty of Medicine, Kagawa University, Kita-gun 761-0793, Japan; 2Department of Internal Medicine, HITO Medical Center, Shikokuchuo 799-0121, Japan; 3Department of Gastroenterology and Hepatology, Takamatsu Red Cross Hospital, Takamatsu 760-0017, Kagawa, Japan; 4Department of Radiology, Kagawa University, Kita-gun 761-0793, Japan

**Keywords:** hepatocellular carcinoma, skeletal muscle, psoas muscle, sarcopenia, hepatic arterial infusion chemotherapy, clinical outcome, prognosis, transcatheter arterial chemoembolization

## Abstract

**Simple Summary:**

Although reduced skeletal muscle mass affects therapeutic efficacy and adverse events in various therapies for hepatocellular carcinoma (HCC), there are no reports on skeletal muscle mass or its changes and prognostic indications in patients with advanced HCC undergoing hepatic arterial infusion chemotherapy (HAIC). This is the first study to show the association between the presence of a decreased skeletal muscle index (SMI) and clinical outcomes in HAIC, with novel evidence having a strong impact. A decrease in the SMI immediately after the start of HAIC was significantly associated with poor progression-free survival and overall survival in patients with advanced HCC, whereas the pre-treatment SMI was not a significant factor. Additionally, patients with a decreased SMI after treatment had worse nutritional status and liver function and poor therapeutic effects. It is effective to monitor the SMI and evaluate the therapeutic effects by using computed tomography to evaluate general conditions and predict clinical outcomes.

**Abstract:**

Sarcopenia-related factors, including the skeletal muscle index (SMI), are reportedly associated with prognosis in patients with hepatocellular carcinoma (HCC) receiving various treatments. However, there is no evidence relating to hepatic arterial infusion chemotherapy (HAIC). In this study, we investigated whether a low SMI was associated with worse clinical outcomes of HAIC. Seventy patients with advanced HCC were included. Clinical outcomes were compared between the decreased SMI (n = 27) and non-decreased SMI (n = 43) groups, which were classified according to changes in the SMI after 3 weeks of treatment. In the prognostic analysis, patients in the decreased SMI group had significantly shorter progression-free and overall survival (OS) than those in the non-decreased SMI group. In addition, poor nutritional status and liver function were associated with an immediate decrease in the SMI after HAIC. The therapeutic effect was worse in the decreased SMI group than in the non-decreased SMI group, although the incidence of adverse events did not significantly differ. In multivariate analysis, a decreased SMI at 3 weeks after HAIC was identified as a significant independent factor associated with OS. A decreased SMI in patients with advanced HCC undergoing HAIC was associated with poor prognosis. It is effective to monitor the SMI to evaluate general conditions and predict clinical outcomes.

## 1. Introduction

Liver cancer is one of the most frequent cancers worldwide, and its incidence continues to rise, with an estimated 1 million cases per year by 2025 [1]. Recent statistics show that liver cancer is the sixth most common cancer and the third most common cause of cancer-related deaths worldwide [2]. Liver cancer has a poor prognosis, with a 5-year survival rate of 18%, and it is the most lethal cancer after pancreatic cancer [3]. Primary liver cancer includes hepatocellular carcinoma (HCC), intrahepatic cholangiocarcinoma, cholangiocarcinoma, mucinous cystadenocarcinoma, and other rare types, such as sarcomas. HCC is the most common histological type, accounting for approximately 90% of cases [1]. Although significant advances have been made in the prevention and diagnosis of HCC in recent years, more than half of patients with HCC are still diagnosed at the intermediate or advanced stage [4], and they are often treated with transcatheter arterial chemoembolization (TACE) and systemic therapy. Major vascular invasion (MVI) and extrahepatic metastasis are the defining factors for advanced-stage HCC, and systemic therapy is recommended [4]. In particular, MVI, such as portal vein tumor thrombus (PVTT), has a poor prognosis because it causes direct hepatic deterioration, and the efficacy of systemic therapy for advanced HCC with MVI remains unsatisfactory [5]. Hepatic arterial infusion chemotherapy (HAIC) is an effective, continuous, and repeated delivery of chemotherapeutic drugs into the hepatic artery, which feeds tumors by using a catheter or subcutaneously implanted reservoir [6]. HAIC is frequently used for conditions that are unlikely to respond to existing treatment options, such as multiple intrahepatic lesions, large lesions, and MVI. Several studies comparing HAIC to conventional therapy, including sorafenib, have reported the efficacy of HAIC in patients with advanced HCC, especially MVI [7,8,9]. More recently, among HAIC regimens, the efficacy of New-FP therapy using fine-powder cisplatin (CDDP) suspended in lipiodol and 5-fluorouracil (5-FU) in MVI- and major PVTT-HCC has been reported, showing high response rates and prolonged survival [10]. We also previously reported that the combination of HAIC with New-FP therapy and radiotherapy has shown favorable outcomes for MVI-HCC, especially when combined with sorafenib post-treatment [11]. Although systemic therapies, including immune checkpoint inhibitors (ICIs) and molecular target agents (MTAs), for advanced HCC have made significant progress and are increasingly being used in clinical practice, HAIC is also a promising treatment option for intrahepatic progressive lesions, especially MVI-HCC, which is difficult to treat with conventional therapies.

Skeletal muscle loss occurs secondarily to a variety of underlying diseases, including liver diseases, renal diseases, malignancy, inflammatory diseases, and sarcopenia, and it is characterized by loss of skeletal muscle mass, strength, and quality [12]. In patients with chronic liver diseases, protein energy malnutrition, inadequate protein synthesis due to branched-chain amino-acid deficiency, decreased testosterone, increased myostatin expression, and increased reactive oxygen species and inflammatory cytokines can contribute to skeletal muscle loss [13]. In fact, skeletal muscle loss has been reported to be closely related to the prognosis of patients with HCC undergoing treatment [14], and there are many reports on the skeletal muscle index (SMI) based on computed tomography (CT), which is one of the diagnostic criteria for sarcopenia normalized by the square of the patient’s height [15,16], because of the ease of follow-up. Several studies on the prognosis of HCC patients undergoing TACE have shown that patients with a pre-treatment SMI below the cutoff have significantly worse overall survival (OS) than those with a normal SMI, and a low SMI is a significant independent factor for shorter OS [17,18]. Studies on systemic therapy for advanced HCC have reported that an SMI below the cutoff is associated with poor clinical outcomes before sorafenib [19,20,21] and lenvatinib treatment [22,23,24]. On the other hand, in patients undergoing HAIC, as well as in those undergoing TACE and systemic therapy, an assessment of their general condition is important in determining the indication for treatment and objective parameters of prognosis. However, there is no evidence on whether skeletal muscle loss or change during treatment is associated with clinical outcomes in patients with HCC undergoing HAIC.

In this study, we examined the prognostic value of skeletal muscle loss in patients with HCC treated with HAIC. The present study aimed to determine whether a decreased SMI immediately after treatment was associated with the prognosis of patients with advanced HCC treated with HAIC. We also examined whether skeletal muscle loss affected nutrient status, liver function, therapeutic effects, and the occurrence of adverse events (AEs) during the clinical course.

## 2. Materials and Methods

### 2.1. Study Design and Protocol

This retrospective cohort study included patients with advanced HCC who were treated with HAIC between January 2009 and January 2022 at Kagawa University. HCC diagnosis was made by using complementary tumor markers such as α-fetoprotein (AFP) and des-γ-carboxy prothrombin (DCP), contrast-enhanced CT, and magnetic resonance imaging (MRI), and if these tests did not show typical HCC findings, a needle biopsy was performed to confirm the diagnosis. To eliminate regimen-specific differences, only patients undergoing HAIC with New-FP therapy were included; patients treated with low-dose 5-FU plus CDDP (LDFP) therapy or miriplatin plus 5-FU (MiF) therapy were excluded. Patients undergoing HAIC with fewer than two courses of New-FP were also excluded because accurate data could not be collected.

### 2.2. Treatment Protocol

HAIC was performed in all patients by inserting an indwelling catheter through the femoral artery, extending the distal end into the hepatic or gastroduodenal artery, and connecting the proximal end to the port system for subcutaneous implantation. The apical or lateral foramen position was adjusted so chemotherapeutic drugs could be infused into the tumors in the liver, and the gastroduodenal, right gastric, and posterior superior pancreaticoduodenal arteries were occluded with microcoils in case of accidental inflow. New-FP therapy was used [25], in which 50 mg of CDDP was suspended in lipiodol and slowly infused under angiography, followed by a 250 mg bolus dose of 5-FU and 1250 mg continuous injection of 5-FU using a balloon pump. A week later, the second course was administered with the same method. After the third course, the dose was reduced to 20–30 mg CDDP and 500–1000 mg 5-FU every 2 weeks until disease progression or unacceptable adverse events occurred. Up to ten courses were administered. Treatment was interrupted as per the recommendations of the manufacturer.

### 2.3. Evaluation of Parameters

General conditions were assessed by using the body mass index (BMI) and Eastern Cooperative Oncology Group performance status (PS) [26]. Nutritional status was assessed by controlling the nutritional status (CONUT) score calculated from peripheral blood lymphocytes, albumin, and cholesterol levels [27]. Liver function was assessed by using the Child–Pugh score, albumin bilirubin (ALBI) score [28], and modified ALBI grade [29]. The clinical stage of HCC was assessed with the tumor–lymph node–metastasis classification based on the criteria of the Liver Cancer Study Group of Japan according to tumor diameter, number of tumors, vascular invasion, lymph node metastasis, and distant metastasis [30].

Contrast-enhanced CT was performed before the start of treatment, 3 weeks later, and every 4 weeks thereafter until the patient died or was lost to follow-up. SYNAPSE VINCENT (Fujifilm, Tokyo, Japan) was used to analyze and measure the muscle and fat areas on the CT images. Muscle mass was assessed by measuring the skeletal muscle and psoas muscle area at the level of the third lumbar vertebra and dividing these areas by the square of the height to calculate the SMI and psoas muscle index (PMI). The cutoff values used to define a low SMI before treatment initiation were 42 cm^2^/m^2^ for males and 38 cm^2^/m^2^ for females according to the guidelines of the Japan Society of Hepatology (JSH) for sarcopenia in liver disease [16]. ΔSMI was calculated as
$$\frac{\mathrm{SMI}\;\mathrm{after}\;\mathrm{treatment}\;-\;\mathrm{SMI}\;\mathrm{before}\;\mathrm{treatment}}{\mathrm{SMI}\;\mathrm{before}\;\mathrm{treatment}}\times 100\;\left(\%\right)$$

Patients with ΔSMI of −10% or less were classified into the decreased SMI group, and others were classified into the non-decreased SMI group. To assess fat mass, the subcutaneous and internal fat areas at the umbilical level were measured. Therapeutic effects were classified as complete response (CR), partial response (PR), stable disease (SD), or progressive disease (PD) according to the modified Response Evaluation Criteria in Solid Tumors (mRESIST) by using contrast-enhanced CT findings [31]. The occurrence of adverse events (AEs) was evaluated according to the Common Terminology Criteria for Adverse Events (CTCAE) version 5.0.

### 2.4. Statistical Analysis

Statistical analyses were performed by using GraphPad Prism (Prism 8.4.3; San Diego, CA, USA). The frequencies between the decreased and non-decreased SMI groups were compared by using the chi-square test or Fisher’s exact test. Continuous variables are presented as the median, and the differences between the decreased and non-decreased SMI groups were compared by using the Mann–Whitney U test. The paired values before and after treatment were compared by using the Wilcoxon signed-rank sum test. Progression-free survival (PFS) and OS rates in January 2023 were calculated by using the Kaplan–Meier method, and significance was determined according to the log-rank test. Multivariate analyses were performed for factors related to PFS and OS by using the Cox proportional hazard model, and significance was determined for each factor by using the Wald test. Statistical significance was set to *p* < 0.05.

### 2.5. Ethical Approval

The present study was approved by the ethics committee of Kagawa University, Faculty of Medicine (Ethics approval 2022-147).

## 3. Results

### 3.1. Patient Characteristics

In this study, 75 patients with advanced HCC underwent HAIC. A flowchart of the patient selection process is shown in Figure 1. Among these, three patients treated with regimens other than New-FP therapy were excluded: two with LDFP therapy and one with MiF therapy. Two patients who received fewer than two courses of New-FP therapy were also excluded. Seventy patients were enrolled and analyzed. The patients were classified according to whether their SMIs decreased by over 10% during the first 3 weeks of treatment, and they were partitioned with 27 patients in the decreased SMI group and 43 patients in the non-decreased SMI group. In addition, 16 patients in the decreased SMI group and 42 patients in the non-decreased SMI group were followed up for over 7 weeks after treatment, allowing additional changes in the SMI to be tracked.

The baseline patient characteristics are shown in Table 1. The median age was 67 (37–87) years in the decreased SMI group and 69 (41–89) years in the non-decreased SMI group, with no significant difference. There were no significant differences in sex, etiology, or BMI between the two groups. The baseline median and percentages below the cutoff of the SMI were 48.25 (31.20–65.05) cm^2^/m^2^ and 81.5% in the decreased SMI group and 46.82 (27.57–61.80) and 76.7% in the non-decreased SMI group, respectively, did they not differ significantly, suggesting that skeletal muscle mass before the start of treatment was not related to subsequent reduction. Notably, more patients in the non-decreased SMI group received MTA therapy after HAIC than those in the decreased SMI group, indicating a higher likelihood of post-treatment. Otherwise, there were no significant differences in the baseline characteristics, including general condition, nutritional status, liver function, and clinical stage.

### 3.2. Skeletal Muscle Mass and Related Factors

The correlations between the baseline SMI and related indicators are shown in Figure 2. Age was negatively correlated with the baseline SMI (Figure 2A), BMI was positively correlated with the baseline SMI (Figure 2B), and there was no correlation between the ALBI score and baseline SMI (Figure 2C). Notably, the PMI was significantly correlated with the baseline SMI (Figure 2D).

Figure 3 shows the comparison of the baseline subcutaneous fat area, internal fat area, and PMI measured from CT images in the decreased and non-decreased SMI groups. There were no significant differences in the subcutaneous and internal fat areas before the start of treatment between the two groups (Figure 3A,B). Post-treatment SMI reduction was not associated with a higher or lower SMI before the start of treatment; similarly, there was no significant difference in the baseline PMI between the two groups (Figure 3C).

### 3.3. Therapeutic Effects

Table 2 shows the best therapeutic effect of mRESIST based on the contrast-enhanced CT findings after HAIC. The overall response rate (CR + PR) was 14.8% in the decreased SMI group and 69.8% in the non-decreased SMI group. The disease control rate (CR + PR + SD) was 55.6% in the decreased SMI group and 88.4% in the non-decreased SMI group. The non-decreased SMI group showed a significantly better therapeutic effect than that of the decreased SMI group.

### 3.4. AEs Due to HAIC

The occurrence of AEs during the clinical course of HAIC is shown in Table 3. AEs of any grade occurred at a rate of 96.3% in the decreased SMI group and at a rate of 79.1% in the non-decreased SMI group. Severe AEs of grade 3 or higher occurred in 51.9% of the decreased SMI group and 39.5% of the non-decreased SMI group. There were no significant differences in AEs depending on whether the SMI decreased during the clinical course of HAIC. The major AEs observed were decreased platelet count, increased transaminase level, anorexia, vomiting, esophageal variceal hemorrhage, decreased white blood cell count, anemia, and upper gastrointestinal hemorrhage. Esophageal variceal hemorrhage was more common in the decreased SMI group than in the non-decreased SMI group; however, other AEs were not significantly different.

### 3.5. Changes in Skeletal Muscle Mass and Related Factors

Figure 4 shows the changes in nutritional status, liver function, and skeletal muscle mass in the 16 patients in the decreased SMI group and 42 patients in the non-decreased SMI group who were followed for at least 7 weeks after the start of HAIC. The CONUT scores increased significantly at 7 weeks after treatment in both the decreased and non-decreased SMI groups compared to the baseline, indicating a possible worsening of nutritional status during treatment (Figure 4A). The ALBI scores were unchanged at 3 weeks after treatment in the decreased SMI group compared to the baseline, but they increased significantly at 7 weeks after treatment, indicating worsening liver function; no significant change was observed in the non-decreased SMI group (Figure 4B). Skeletal muscle mass showed a significant decrease in the SMI in the decreased SMI group at 7 weeks, as well as at 3 weeks after treatment, compared to the baseline, while there were no significant changes in the SMI in the non-decreased SMI group at 7 weeks after treatment (Figure 4C). Examination of the psoas muscle mass showed a significant decrease in the PMI at 3 and 7 weeks after treatment in the decreased SMI group compared to the baseline (Figure 4D). The SMI and PMI showed similar trends during the clinical course of HAIC.

### 3.6. Prognosis Analysis

The clinical outcomes are shown in Figure 5, and the median duration of observation for censored cases was 8.4 (1.0–102.0) months. The median PFS was 2.2 (0.2–4.5) months in the decreased SMI group and 5.0 (0.5–58.4) months in the non-decreased SMI group, which was a significant difference (Figure 5A). The median OS was 3.8 (1.0–8.4) months in the decreased SMI group and 19.5 (1.5–102.0) months in the non-decreased SMI group (Figure 5B). The Kaplan–Meier survival curves for both groups showed a significant difference with a hazard ratio (HR) of 4.60 (95% CI 1.85–11.4) (*p* < 0.01).

We analyzed the PFS and OS in patients with advanced HCC after HAIC. In the multivariate analyses, the independent factors significantly associated with poor PFS were a decreased SMI after 3 weeks of HAIC (yes vs. no: HR 5.59; 95% CI, 3.00–10.69; *p* < 0.01) and a Child–Pugh score of 7 or higher (≥7 vs. ≤6: HR 2.21; 95% CI, 1.25–3.82; *p* < 0.01) (Table 4). Furthermore, the independent factors significantly associated with poor OS were a decreased SMI after 3 weeks of HAIC (yes vs. no: HR 13.10; 95% CI, 4.91–39.70; *p* < 0.01) in the multivariate analyses (Table 5).

## 4. Discussion

HAIC is a promising treatment option for advanced HCC that is unlikely to respond to TACE or systemic therapies, such as in cases with multiple intrahepatic lesions, large intrahepatic lesions, and MVI. The efficacy of HAIC in advanced HCC, especially MVI, has been reported [7,8,9]. To the best of our knowledge, the present study is the first to show that a decreased SMI immediately after treatment is associated with the prognosis of patients with advanced HCC treated with HAIC.

Treatment of HCC includes liver resection, local ablative therapy, TACE, HAIC, and systemic therapies—and even combinations of these therapies—and a multidisciplinary approach is essential in the selection of treatment, as it is important to understand the patient’s systemic status. Although nutritional status and skeletal muscle mass are clearly related to the outcomes of patients with HCC, current staging and prognostic systems do not include these parameters. An imbalance between anabolic and catabolic metabolism due to malnutrition, decreased activity, increased inflammatory response, and decreased testosterone levels results in abnormal skeletal muscle mass in patients with HCC [32], suggesting that skeletal muscle loss has a negative impact on prognosis in patients who undergo various treatments. According to reports on radical treatment, a low pre-treatment SMI was shown to be a significant independent risk factor for recurrence in non-obese patients with HCC undergoing hepatic resection [33]. Another meta-analysis showed that in patients with HCC undergoing liver resection, a low pre-operative SMI was significantly associated with mortality [34]. A study on radiofrequency ablation (RFA), the recent mainstay of local ablative therapy, showed that a low SMI was significantly associated with recurrence and mortality in patients with early-stage HCC who had undergone RFA [35]. In addition, a cohort study by another group showed that a low PMI was a prognostic factor that worsened the 5-year OS rate for patients with early-stage HCC who underwent RFA [36].

When HCC progresses to the unresectable or metastatic stages, the indication for treatment is limited to systemic therapy, and it is important to evaluate the patient’s general condition to maximize the therapeutic effects. In patients with advanced HCC, the association between clinical outcomes and skeletal muscle loss with systemic therapies has also been widely reported, most frequently with sorafenib and lenvatinib. Several recent studies reported that a CT-based SMI below the cutoff was associated with poor clinical outcomes, including PFS and OS, in patients with advanced HCC treated with sorafenib [19,20,21] and lenvatinib [22,23,24]. Interestingly, in studies focusing on the decline in the SMI after treatment, HCC patients with a decreased CT-L3 SMI after sorafenib treatment had significantly shorter survival, suggesting that rapid skeletal muscle mass loss is associated with poor prognosis [37]. In a study on lenvatinib treatment, HCC patients with a decreased CT-L3 PMI had significantly worse OS in the PMI severe wasting group than in the PMI mild wasting group, and there was a correlation between a PMI with severe wasting and OS [38]. Despite the increasing number reports on MTAs, such as sorafenib and lenvatinib, there is no evidence for an association between clinical outcomes and skeletal muscle loss in HAIC for advanced HCC.

In this study, 27 patients (38.6%) had a decrease in their SMIs after 3 weeks of HAIC, but there were no significant differences in the median baseline SMI in the decreased SMI and the non-decreased SMI groups, indicating that there was no association between the presence of a decrease in the SMI after treatment and the baseline SMI values. Furthermore, there were no significant differences in several clinical characteristics, such as baseline nutritional status, liver function, degree of tumor progression, and fat-area-based CT images, between these groups. In the prognostic analysis, patients in the decreased SMI group had significantly shorter PFS and OS than those in the non-decreased SMI group. Interestingly, patients in the decreased SMI group had significantly increased ALBI and CONUT scores after 7 weeks of HAIC, suggesting that in addition to poor nutritional status, poor liver function could be associated with an immediate decrease in skeletal muscle mass after HAIC. Treatment to maintain nutritional status and liver function, such as branched-chain amino acid (BCAA) supplementation, may be a promising treatment strategy for improving outcomes in HCC patients treated with HAIC. Another factor related to poor prognosis was that the best therapeutic effect was worse in the decreased SMI group than that in the non-decreased SMI group, although the incidence of AEs was not significantly different. In multivariate analysis, a decreased SMI at 3 weeks after HAIC was identified as a significant and strong independent factor associated with OS, whereas the pretreatment SMI was not a significant factor. Other authors have also shown that the SMI before HAIC is not a significant prognostic factor [39], but, unlike patients who are eligible for resection or RFA, patients who undergo HAIC may have a very poor prognosis of only a few months of life. Thus, these patients may be less susceptible to the baseline SMI and overall condition because of the shorter observation period. As this study has shown, it makes sense to focus on the change in, rather than the baseline, SMI in poor-prognosis patients who undergo HAIC, since worsening nutritional status and liver function resulting from poor therapeutic response are directly associated with immediate SMI decline. In fact, post-treatment parameters and biomarkers in general, not only the SMI, are rarely examined as prognostic factors in HAIC. Transient transaminase elevation after TACE has been reported to be a predictor of the objective response rate [40]. Additionally, liver function tested directly after HAIC, along with the monitoring of changes in the SMI, may be clinically useful. Furthermore, the PMI may be another more convenient indicator of muscle mass. It was previously reported that the SMI and PMI were correlated in healthy adults. The present study also showed that the baseline SMI and PMI were positively correlated in patients with advanced HCC and that the PMI similarly decreased in patients whose SMI decreased over the course of HAIC treatment [41]. The PMI can be measured from CT images with simple methods, such as the manual trace method, without dedicated muscle measurement software, making it a more frequently used parameter for assessing general conditions and predicting prognosis in patients with advanced HCC treated with HAIC.

Our study had some limitations. First, this was a retrospective single-center study, and the number of enrolled patients was small. In addition, the small sample size may have made it difficult to properly compare the occurrence of AEs with HAIC between the decreased SMI and non-decreased SMI groups. HAIC is a treatment technique with many differences between centers, and to eliminate technical bias due to the regimen used, this study was limited to the New-FP treatment at a single center. Further, recently, the number of patients with HCC undergoing HAIC has decreased due to the widespread use of ICIs and/or MTAs for advanced HCC, and patient recruitment for this study may have been affected by the time. Second, there was a lack of consideration of muscle strength—that is, grip strength and walking speed—which is part of the definition of sarcopenia. Third, although BCAAs and levocarnitine are effective in preventing muscle loss, we could not analyze the effects of BCAAs and levocarnitine because only a few patients were supplemented with them. Despite these limitations, this study is the first to show that a decreased SMI immediately after treatment is associated with prognosis in patients with HCC on HAIC and can be novel evidence of a strong clinical impact.

## 5. Conclusions

A decreased SMI in patients with advanced HCC undergoing HAIC is associated with poor prognosis with worsening nutritional status, decreased liver function, and worsened therapeutic effects. It is effective to monitor skeletal muscle mass and evaluate the therapeutic effects of CT in order to evaluate the general condition and predict clinical outcomes.

## Figures and Tables

**Figure 1 cancers-15-01834-f001:**
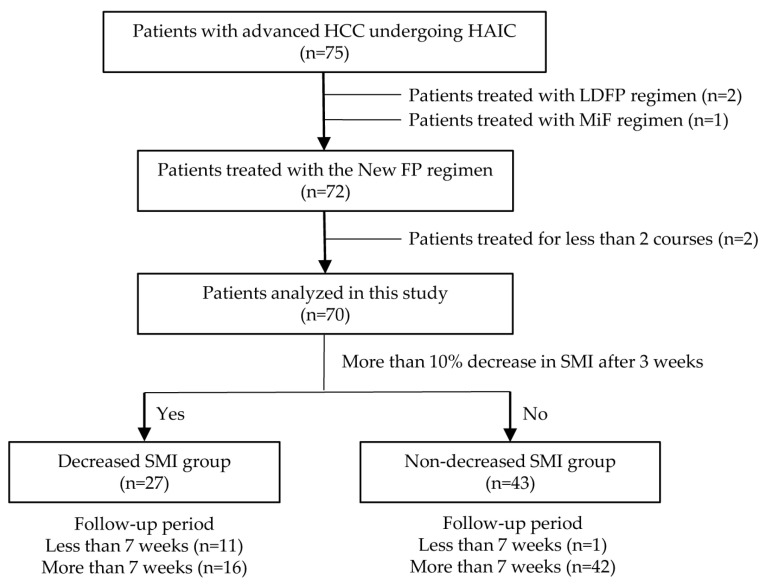
Flowchart of patient selection. Initially, there were 74 eligible patients with hepatocellular carcinoma (HCC) treated with hepatic arterial infusion chemotherapy (HAIC). Two patients who received low-dose 5-FU plus CDDP (LDFP) therapy and one who received miriplatin plus 5-FU (MiF) were excluded, and only patients who received New-FP therapy were selected. Patients who received less than two courses of New-FP therapy were also excluded. A total of 70 patients were finally enrolled in this study. The patients were classified according to whether their skeletal muscle index (SMI) decreased by more than 10% during the first 3 weeks of treatment and were divided into 27 patients in the decreased SMI group and 43 patients in the non-decreased SMI group. Furthermore, 16 patients in the decreased SMI group and 42 patients in the non-decreased SMI group could be followed up for more than 7 weeks.

**Figure 2 cancers-15-01834-f002:**
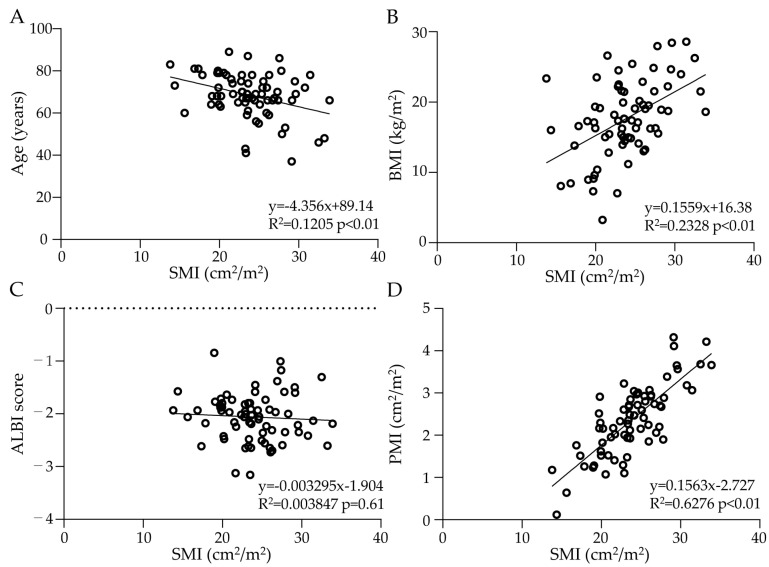
The correlation between the baseline skeletal muscle index (SMI) and related factors. (**A**) Age was negatively correlated with the baseline SMI. (**B**) Body mass index (BMI) was positively correlated with the baseline SMI. (**C**) Albumin bilirubin (ALBI) scores were not correlated with the baseline SMI. (**D**) The psoas muscle index (PMI) in the baseline was positively correlated with the baseline SMI.

**Figure 3 cancers-15-01834-f003:**
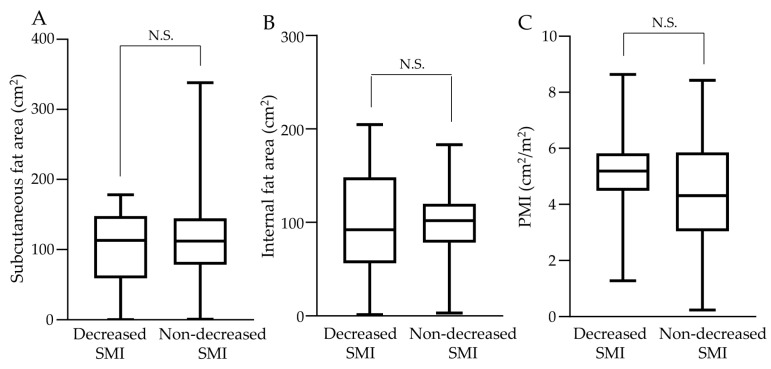
The comparison of fat area and muscle area in the CT findings before the start of treatment. (**A**) The median subcutaneous fat area at the umbilical level was 113.26 (0.02–178.14) cm^2^ in the decreased skeletal muscle index (SMI) group and 112.26 (0.9–338) cm^2^ in the non-decreased SMI group; these were not significantly different. (**B**) The median internal fat area at the umbilical level was 92.22 (1.48–204.50) cm^2^ in the decreased SMI group and 101.93 (3.23–183.11) cm^2^ in the non-decreased SMI group; these were not significantly different. (**C**) The median PMI was 5.17 (1.28–8.64) in the decreased SMI group and 4.31 (0.24–8.43) cm^2^/m^2^ in the non-decreased SMI group; these were not significantly different. N.S.; not significant.

**Figure 4 cancers-15-01834-f004:**
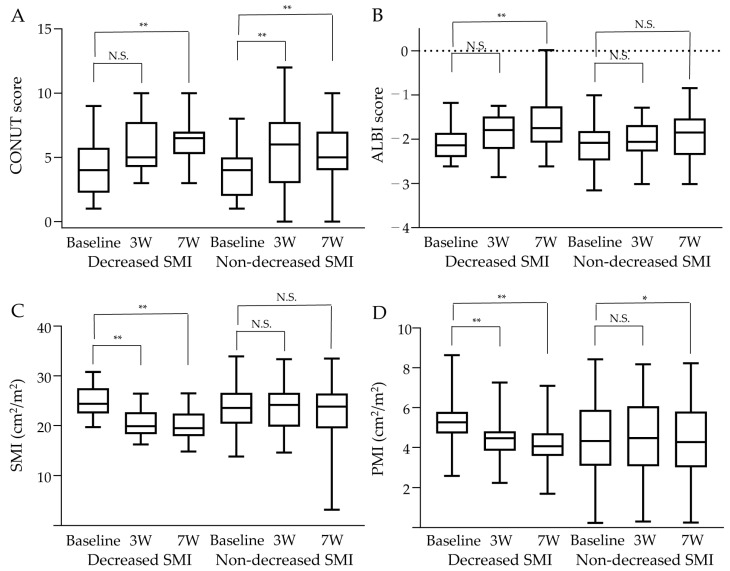
The changes in skeletal muscle mass and related factors in the members of decreased SMI (n = 16) and non-decreased SMI groups (n = 42) who were followed up for at least 7 weeks after the start of hepatic artery infusion chemotherapy (HAIC). (**A**) Controlling nutritional status (CONUT) scores increased significantly at 7 weeks after treatment in both the decreased skeletal muscle index (SMI) group and the non-decreased SMI group compared to the baseline. (**B**) The albumin bilirubin (ALBI) scores were unchanged at 3 weeks after treatment in the decreased SMI group compared to the baseline, but they increased significantly at 7 weeks after treatment. No significant changes were observed in the non-decreased SMI group. (**C**) The SMI was significantly decreased at 7 weeks, as well as at 3 weeks after treatment, compared to the baseline in the decreased SMI group. There were no significant changes in the SMI in the non-decreased SMI group at 7 weeks after treatment. (**D**) The psoas muscle index (PMI) was significantly decreased at 3 and 7 weeks after treatment in the decreased SMI group compared to the baseline. * *p* < 0.05 and ** *p* < 0.01. N.S.; not significant, 3W; 3 weeks, 7W; 7 weeks.

**Figure 5 cancers-15-01834-f005:**
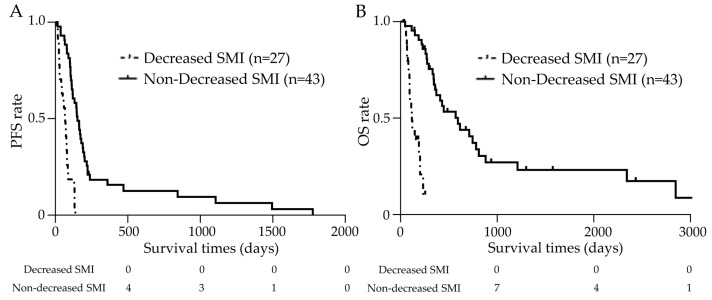
Results of the prognostic analysis using the Kaplan–Meier curves. (**A**) The median progression-free survival (PFS) was 2.2 (0.2–4.5) months in the decreased skeletal muscle index (SMI) group and 5.0 (0.5–58.4) months in the non-decreased SMI group (*p* < 0.01). (**B**) The median overall survival (OS) was 3.8 (1.0–8.4) months in the decreased SMI group and 19.5 (1.5–102.0) months in the non-decreased SMI group (*p* < 0.01).

**Table 1 cancers-15-01834-t001:** The baseline patient characteristics.

	Decreased SMI	Non-Decreased SMI	
Characteristics	n = 27	n = 43	*p* Value
Age (years): median (range)	67 (37–87)	69 (41–89)	0.38
Sex (n): Male/Female (%)	24 (88.9)/3 (11.1)	33 (76.7)/10 (23.3)	0.34
Etiology (n): HBV/HCV/Both/NBNC (%)	6 (22.2)/14 (51.9)/1 (3.7)/6 (22.2)	8 (18.6)/25 (58.1)/0 (0.0)/10 (23.3)	0.61
Height (m): median (range)	1.63 (1.44–1.82)	1.61 (1.34–1.71)	0.30
Body weight (kg): median (range)	61.5 (47.4–77.0)	61.3 (40.4–81.5)	0.56
BMI (kg/m^2^): median (range)	23.6 (18.5–28.1)	23.8 (16.6–29.3)	0.77
Baseline SMI (kg/m^2^): median (range)	Male: 49.11 (31.20–65.05) Female: 46.56 (37.93–47.20)	Male: 48.93 (28.72–67.80) Female: 41.15 (27.57–49.25)	0.47
Above SMI cutoff (n): Yes/No (%)	22 (81.5)/5 (18.5)	33 (76.7)/10 (23.3)	0.77
Performance status (n): 0/1/2/3 (%)	14 (51.9)/8 (29.6)/4 (14.8)/1 (3.7)	32 (74.4)/9 (20.9)/2 (4.7)/0 (0.0)	0.15
CONUT score (n): 0–1/2–4/5–8/ >8 (%)	1 (3.7)/13 (48.1)/10 (37.0)/3 (11.1)	4 (9.3)/24 (55.8)/15 (34.9)/0 (0.0)	0.13
Recurrence (n): Yes/No (%)	8 (29.6)/19 (70.4)	16 (37.2)/27 (62.8)	0.43
Combined radiotherapy for MVI (n): Yes/No (%)	15 (55.6)/12 (44.4)	30 (69.8)/13 (30.2)	0.23
Post-MTA treatment (n): Yes/No (%)	8 (29.6)/19 (70.4)	28 (65.1)/15 (34.9)	<0.01
Child–Pugh score (n): 5/6/7/≥8 (%)	8 (29.6)/6 (22.2)/8 (29.6)/5 (18.5)	15 (34.9)/19 (44.2)/5 (11.6)/4 (9.3)	0.10
mALBI grade (n): 1/2a/2b/3 (%)	1 (3.7)/6 (22.2)/17 (63.0)/3 (11.1)	9 (20.9)/5 (11.6)/27 (62.8)/2 (4.7)	0.13
AFP (ng/mL): median (range)	3528 (3–202,847)	1942 (2–337,408)	0.07
DCP (mAU/mL): median (range)	12,346 (13–901,588)	4348 (9–1,114,735)	0.19
Maximum tumor size (n): <3 cm/≥3 cm (%)	2 (7.4)/25 (18.5)	7 (16.3)/36 (83.7)	0.47
Number of tumors (n): ≤3/≥4 (%)	1 (3.7)/26 (96.3)	7 (16.3)/36 (83.7)	0.14
Major vascular invasion (n): Yes/No (%)	23 (85.2)/4 (14.8)	31 (72.1)/12 (27.9)	0.25
Extrahepatic metastasis (n): Yes/No (%)	5 (18.5)/22 (81.5)	5 (11.6)/38 (88.4)	0.49
TMN staging LCSGJ 6th (n): III/IVa/IVb (%)	3 (11.1)/19 (70.4)/5 (18.5)	13 (30.2)/26 (60.5)/4 (9.3)	0.14

HBV, hepatitis B virus; HCV, hepatitis C virus; NBNC, non-HBV and non-HCV; BMI, body mass index; SMI, skeletal muscle mass; CONUT, controlling nutritional status; MVI, major vascular invasion; MTAs, molecular target agents; mALBI, modified albumin bilirubin; AFP, α-fetoprotein; DCP, des-γ-carboxy prothrombin; TNM, tumor–lymph node metastasis; LCSGJ; Liver Cancer Study Group of Japan.

**Table 2 cancers-15-01834-t002:** The best therapeutic effect as assessed with modified RESIST.

	Decreased SMI	Non-Decreased SMI	
Therapeutic Effect	n = 27	n = 43	*p* Value
ORR, n (%)	4 (14.8)	30 (69.8)	<0.01
DCR, n (%)	15 (55.6)	38 (88.4)	<0.01
CR, n (%)	0 (0.0)	7 (16.3)	
PR, n (%)	4 (14.8)	23 (53.5)	
SD, n (%)	11 (40.7)	8 (18.6)	
PD, n (%)	12 (44.4)	5 (11.6)	

SMI, skeletal muscle index; ORR, overall response rate; DCR, disease control rate; CR, complete response; PR, partial response; SD, stable disease; PD, progressive disease.

**Table 3 cancers-15-01834-t003:** AEs that occurred during the clinical course of HAIC.

	Decreased SMI	Non-Decreased SMI	
	n = 27	n = 43	*p* Value
Any adverse events, n (%)	26 (96.3)	34 (79.1)	0.08
Grade 1	3 (11.1)	3 (6.98)	
Grade 2	9 (33.3)	14 (32.6)	
Grade 3	9 (33.3)	16 (37.2)	
Grade 4	5 (18.5)	1 (2.3)	
Grade 5	0 (0.0)	0 (0.0)	
Major adverse events, n (%)			
Platelet count decreased	11 (40.7)	25 (58.1)	0.16
AST increased	9 (33.3)	9 (20.9)	0.25
Anorexia	8 (29.6)	6 (14.0)	0.11
Vomiting	5 (18.5)	2 (4.7)	0.10
Esophageal variceal hemorrhage	4 (14.8)	0 (0.0)	<0.05
WBCs decreased	2 (7.4)	4 (9.3)	>0.99
Anemia	2 (7.4)	7 (16.3)	0.47
Upper gastrointestinal hemorrhage	0 (0.0)	3 (7.0)	0.28
Severe adverse events of grade ≥ 3, n (%)	14 (51.9)	17 (39.5)	0.31
Platelet count decreased	4 (14.8)	9 (20.9)	
Anorexia	3 (11.1)	0 (0.0)	
Esophageal variceal hemorrhage	3 (11.1)	0 (0.0)	
Diarrhea	3 (11.1)	0 (0.0)	
WBCs decreased	2 (7.4)	3 (7.0)	
AST increased	2 (7.4)	0 (0.0)	
Wound infection	2 (7.4)	0 (0.0)	
Anemia	1 (3.7)	3 (7.0)	
Vomiting	1 (3.7)	1 (2.3)	
Tumor lysis syndrome	1 (3.7)	0 (0.0)	
Urinary tract infection	1 (3.7)	0 (0.0)	
Cholecystitis	1 (3.7)	0 (0.0)	
Creatinine increased	1 (3.7)	0 (0.0)	
Hepatic failure	1 (3.7)	0 (0.0)	
Upper gastrointestinal hemorrhage	0 (0.0)	3 (7.0)	
Fracture	0 (0.0)	1 (2.3)	
Hepatic infection	0 (0.0)	1 (2.3)	

SMI, skeletal muscle index; AST, aspartate aminotransferase; WBC, white blood cell.

**Table 4 cancers-15-01834-t004:** Univariate and multivariate analyses of progression-free survival.

Variable	n = 70	Univariate Analysis			Multivariate Analysis		
		HR	95%CI	*p* Value	HR	95%CI	*p* Value
Sex	Male	1.56	0.86–3.07	0.17			
Age	≥75 years	0.83	0.48–1.38	0.48			
BMI	<23 kg/m^2^	0.73	0.42–1.23	0.24			
Baseline SMI	Male <42 cm^2^/m^2^ or Female <38 cm^2^/m^2^	0.61	0.32–1.09	0.11			
Decreased SMI at 3 weeks	≥10%	5.65	3.06–10.66	<0.01	5.59	3.00–10.69	<0.01
PS	≥1	1.23	0.71–2.05	0.44			
CONUT score	≥2	1.81	0.79–5.21	0.21			
Etiology	HBV	1.36	0.74–2.39	0.30			
Previous treatment	Yes	0.81	0.50–1.32	0.39			
Child–Pugh score	≥7	2.33	1.34–3.95	<0.01	2.21	1.25–3.82	<0.01
mALBI grade	2a, 2b, or 3	1.21	0.63–2.63	0.59			
AFP	>300 ng/mL	1.12	0.67–1.95	0.69			
DCP	>700 mAU/mL	1.30	0.78–2.27	0.33			
Maximum size of tumor	≥50 mm	1.66	1.00–2.84	0.06			
Number of tumors	≥4	2.03	0.97–4.96	0.09			
Major vascular invasion	Yes	0.63	0.36–1.16	0.12			
Extrahepatic metastasis	Yes	1.01	0.48–1.91	0.98			

BMI, body mass index; SMI, skeletal muscle index; PS, performance status; CONUT, controlling nutritional status; mALBI, modified albumin bilirubin; AFP, α-fetoprotein; DCP, des-γ-carboxy prothrombin.

**Table 5 cancers-15-01834-t005:** Univariate and multivariate analyses of overall survival.

Variable	n = 70	Univariate Analysis			Multivariate Analysis		
		HR	95%CI	*p* Value	HR	95%CI	*p* Value
Sex	Male	1.10	0.54–2.56	0.80			
Age	≥70 years	0.50	0.25–0.95	<0.05	0.98	0.95–1.01	0.17
BMI	<23 kg/m^2^	0.73	0.38–1.35	0.33			
Baseline SMI	Male <42 cm^2^/m^2^ or Female <38 cm^2^/m^2^	0.40	0.15–1.01	0.06			
Decreased SMI at 3 weeks	≥10%	12.86	5.00–37.78	<0.01	13.10	4.91–39.70	<0.01
PS	≥1	1.37	0.71–2.53	0.32			
CONUT score	≥5	1.53	0.83–2.77	0.16			
Etiology	NBNC	1.48	0.73–2.83	0.25			
Previous treatment	Yes	0.92	0.51–1.68	0.79			
Child–Pugh score	≥7	1.93	0.99–3.60	<0.05	1.86	0.93–3.54	0.07
mALBI grade	2a, 2b, or 3	2.57	1.03–8.58	0.07			
AFP	>300 ng/mL	1.24	0.66–2.52	0.52			
DCP	>700 mAU/mL	1.83	0.96–3.79	0.08			
Maximum size of tumor	≥50 mm	1.70	0.93–3.21	0.09			
Number of tumors	≥4	1.93	0.82–5.68	0.17			
Major vascular invasion	Yes	1.82	0.86–4.48	0.15			
Extrahepatic metastasis	Yes	0.83	0.31–1.85	0.68			

BMI, body mass index; SMI, skeletal muscle index; PS, performance status; CONUT, controlling nutritional status; mALBI, modified albumin bilirubin; AFP, α-fetoprotein; DCP, des-γ-carboxy prothrombin.

## Data Availability

The data that support the findings of this study are not publicly available because they contain information that could compromise the privacy of the research participants, but they are available from the corresponding author (K.O.) upon reasonable request.

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
