# Peer review of "Prognostic Value of Skeletal Muscle Loss in Patients with Hepatocellular Carcinoma Treated with Hepatic Arterial Infusion Chemotherapy"

_cancers, 2023, doi:10.3390/cancers15061834_

Round 1
Reviewer 1 Report
The authors investigated the impact of the change in the SMI three weeks after the induction of HAIC as a prognostic factor. HAIC is characterized by high response rate and applied for the patients with advanced stage HCC. However, development of the recent immune therapy limits indication of HAIC, it is important the analysis of the prognostic factor about case with efficacy.
1. It has been reported that a skeletal muscle is a significant prognostic factor in various treatment for HCC. In most reports, baseline SMI stratifies prognosis. However, the baseline SMI is not a prognostic factor in HAIC. Further discussion is needed about the reason.
2. There is no description of PMI in Table 1. PMI appears by the last prognostic factor analysis mainly. Both PMI and SMI are assessment of the similar muscle, these factors have a strong correlation, shown in Fig 2. These two factors are strong confounders, the authors should select only SMI for an explanation variable.
3. The criteria of the SMI are different in men and women. It is necessary to show it according to sex. Please describe by men and women in Table 1.
Author Response
Response to reviewer 1 comments
- It has been reported that a skeletal muscle is a significant prognostic factor in various treatment for HCC. In most reports, baseline SMI stratifies prognosis. However, the baseline SMI is not a prognostic factor in HAIC. Further discussion is needed about the reason.
Response: Thank you very much for your comments.
In this study, multivariate analysis identified decrease in SMI at 3 weeks after HAIC as a significant and strong independent factor associated OS, whereas pretreatment SMI was not a significant factor. Other authors have also shown that baseline SMI before HAIC is not significant prognostic factor (PLoS One. 14(6), e021836, 2019). One possible reason why baseline SMI is not a prognostic factor for HAIC is that, unlike patients for whom resection or RFA is indicated, patients undergoing HAIC have a very poor prognosis with a few months of life, and may be less susceptible to baseline muscle mass and general condition due to the short observation period. As this study has shown, it makes sense to focus on the change in SMI, rather than baseline, in poor prognosis patients who undergo HAIC, since worsening nutritional status and liver function resulting from poor therapeutic response are sensitively associated with immediate SMI decline.
As you pointed out, we have revised the corresponding part of discussion section.
- There is no description of PMI in Table 1. PMI appears by the last prognostic factor analysis mainly. Both PMI and SMI are assessment of the similar muscle, these factors have a strong correlation, shown in Fig 2. These two factors are strong confounders, the authors should select only SMI for an explanation variable.
Response: Thank you very much for your comments.
As you pointed out, PMI calculated from CT area at the third lumber level is useful for the assessment of muscle mass, and it has been reported that PMI correlates with SMI measured by bioelectrical impedance analysis (BIA) and CT findings (Minerva Gastroenterol Dietol. 59(2), 173-186, 2013). PMI is versatile in actual clinical practice because it can be measured simply without the use of dedicated muscle mass measurement software.
As shown in Figure 2 of this paper, PMI was positively correlated with SMI in advanced HCC patients who underwent HAIC in our study as well.
We agree with your opinion that PMI and SMI are similar muscle measures and strong confounders, so we have reanalyzed the multivariate analysis on PFS and OS excluding PMI as an explanation variable. The results of the reanalysis identified decreased SMI at 3 weeks after HAIC as a significant independent factor associated with shorter PFS and OS.
To reflect these changes, we have revised the corresponding part of the results section, Table 4, and Table 5.
- The criteria of the SMI are different in men and women. It is necessary to show it according to sex. Please describe by men and women in Table 1.
Response: Thank you very much for your comments.
As you point out, skeletal muscle mass varies by gender and race, so cutoff values used to determine sarcopenia are also different. For example, the Japan Society of Hepatology (JSH) for sarcopenia in liver disease defines the cutoff value of SMI calculated from the CT area at the third lumber level as 42 cm2/m2 for male and 38 cm2/m2 for female (Hepatol Res. 51(9), 957-967, 2021).
In this study, the median baseline SMI was 49.11 (31.20-65.05) cm2/m2 for male and 46.56 (37.93-47.20) cm2/m2 for female in the decreased SMI group and 48.93 (28.72-67.80) cm2/m2 for male and 41.15 (27.57-49.25) cm2/m2 for female in the non-decreased SMI group; SMI was different by gender.
We agree that this point of yours is important, so we have revised Table 1 to include median SMI separately for males and females.

Reviewer 2 Report
It would be interesting to compare the SMI with other sarcopenic measures. Could the authors at least comment on this?
Sample size is limited and this should be commented as a limitation to the study
How were cut-off measures obtained in regression analysis?
The authors should analyze or at least comment the prognostic role of hypertransaminasemia after the loco-regional treatment (cite the recent paper PMID: 34683182)
English grammar should be improved
Author Response
Response to reviewer 2 comments
- It would be interesting to compare the SMI with other sarcopenic measures. Could the authors at least comment on this?
Response: Thank you very much for your comments.
As shown in Figure 2 of our manuscript, age was negatively correlated with baseline SMI before HAIC, and BMI was positively correlated with baseline SMI. Alternatively, SMI did not correlate with ALBI score, which indicates liver function, or CONUT score, which indicates nutritional status. However, in Figure 4, ALBI score was significantly more likely to decrease in the decreased SMI group, where SMI decreased immediately after HAIC, indicating that it is important to focus on post-treatment change for these indicators.
Notably, in Figure 2D, baseline psoas muscle index (PMI) was shown to correlate with baseline SMI. PMI calculated from CT area at the third lumber level is useful for the assessment of muscle mass, and it has been reported that PMI correlates with SMI measured by bioelectrical impedance analysis (BIA) and CT findings (Minerva Gastroenterol Dietol. 59(2), 173-186, 2013). PMI is versatile in actual clinical practice because it can be measured simply without the use of dedicated muscle mass measurement software.
Sarcopenia is characterized by loss of muscle strength and decreased muscle mass and occurs secondary to various underlying diseases such as liver, renal and inflammatory diseases and malignancies (Age Ageing. 43(6), 748-759, 2014). Since this study was conducted retrospectively, it lacks data on muscle strength, one of the definitions of sarcopenia, i.e., grip strength and walking speed, and this included in the manuscript as a limitation. Despite this limitation, this study is the first to show that decreased SMI immediately after treatment is associated with prognosis in patients with HCC on HAIC and can be novel evidence of a strong clinical impact.
Thank you for your kind comment.
- Sample size is limited and this should be commented as a limitation to the study.
Response: Thank you very much for your comments.
HAIC is a treatment technique with many differences depending on the institution where it is performed and the regimen used. Although future studies with large sample sizes including multicenter cases are need, this study was designed to examine a limited sample size at a single institution in a retrospective manner to eliminate technical bias.
We agree with your opinion. Therefore, we have added this point as a limitation in the discussion section.
- How were cut-off measures obtained in regression analysis?
Response: Thank you very much for your comments.
For the determination of cutoff values for explanatory variables in the Cox hazard proportional model, if there were validated values in actual clinical practice, those values were used; otherwise, ROC analysis was used to calculate the cutoffs.
Regarding age, most developed countries accept the chronological age of 65 years as the definition of an elderly person. However, many clinical studies about HCC in the elderly population defined elderly as over 75 years of age (World J Gastroenterol. 25(27), 3563-3571, 2019), and we have adopted 75 years as cutoff values. Regarding BMI, WHO has an international standard of BMI 25 kg/m2 or higher for overweight and 30 kg/m2 or higher as obese, but Asians have a higher risk of cancer and death even at BMI 25 kg/m2 or lower, so the same standard cannot necessarily be used. In fact, a pooled analysis of cohort studies for the Japanese population, considering BMI 23-25 kg/m2 as the criterion, has shown that the other categories have a higher risk of death including cancer (Journal of Epidemiology. 21, 417-430, 2011). Since the median BMI in Japanese population is 23 kg/m2 and the median BMI in the participants in this study was also 23 kg/m2, we used BMI 23 kg/m2 as the cutoff value. Regarding tumor makers including AFP and DCP, ROC analysis was used to calculate the cutoffs. Regarding baseline SMI, the Japan Society of Hepatology (JSH) for sarcopenia in liver disease definite SMI <42 cm2/m2 for male and SMI <38 cm2/m2 for female as diagnostic criteria for sarcopenia (Hepatol Res. 51(9), 957-967, 2021), and these SMI values were also used as cutoff values in this study. In addition, SMI decrease was defined as less than 10% decrease using time-dependent ROC analysis.
- The authors should analyze or at least comment the prognostic role of hypertransaminasemia after the loco-regional treatment (cite the recent paper PMID: 34683182).
Response: Thank you very much for your comments.
As this study has shown, it makes sense to focus on the change in SMI, rather than baseline, in poor prognosis patients who undergo HAIC, since worsening nutritional status and liver function resulting from poor therapeutic response are sensitively associated with immediate SMI decline. As matter of fact, post-treatment parameters and biomarkers are rarely examined as prognostic factors in HAIC, not only SMI; transient transaminase elevation after TACE has been reported to be a predictor of objective response rate (J Pers Med. 11(10), 2021), and liver function performed in the few days after HAIC, along with monitoring changes in SMI, may be clinically useful in the future.
As you pointed out, we have revised the corresponding part of discussion section.
- English grammar should be improved
Response: Thank you very much for your comments.
According to your comment, we have asked a proofreading company to correct our manuscript. Moreover, we have checked the entire manuscript for possible errors and improved the presentation.

Reviewer 3 Report
In this study, the authors investigated the prognostic value of skeletal muscle loss in patients with hepatocellular carcinoma (HCC) treated with hepatic arterial infusion chemotherapy (HAIC), and examined whether skeletal muscle loss affects nutrient status, liver function, therapeutic effects, and the occurrence of adverse events (AEs) during the clinical course. The authors concluded that decreased skeletal muscle index (SMI) in patients with advanced HCC undergoing HAIC is associated with poor prognosis with worsening nutritional status, decreased liver function, and worsened therapeutic effects.
Comments:
The reviewer has some concerns as follows:
1. One of the major concerns is that sample size is really not high, especially only n=27 in decreased SMI group with follow up period n=11 in < 7 week and n=16 in > 7 week. The small number of samples would affect the reliability of the results, the authors should give a reasonable explanation.
2. This retrospective cohort study recruited patients between January 2009 and January 2022. How has COVID-19 affected patient recruitment during the last three years of this period? This issue should be explained and discussed.
3. In the Methods-2.3. Evaluation of Parameters, the authors described “Muscle mass was assessed by measuring the skeletal muscle and psoas muscle area at the level of the third lumbar vertebra and dividing these areas by the square of the height, SMI, and psoas muscle index (PMI)’. The so-called “skeletal muscle” here refers to what parts are measured? Does it contain psoas muscle? Why is the psoas muscle part measured in addition? These issues should be clarified.
4. In Table 3, the results showed that the number of samples was too small to make the AEs results unreliable. The authors should give a reasonable explanation.
5. In Figure 4, the meanings for star symbols (* and **) should be clearly described.
6. In Tables 4 and 5, what are the recruited patient numbers?
7. In Abstract, line 26, “skeletal muscle mass (SMI)” can be revised.
Author Response
Response to reviewer 3 comments
- One of the major concerns is that sample size is really not high, especially only n=27 in decreased SMI group with follow up period n=11 in < 7 week and n=16 in > 7 week. The small number of samples would affect the reliability of the results, the authors should give a reasonable explanation.
Response: Thank you very much for your comments.
HAIC is a treatment technique with many differences between institutions, and to eliminate technical bias due to the regimen used, this study was limited to New-FP treatment at a single center. In addition, in recent year, the number of patients with HCC undergoing HAIC has decreased due to the widespread use of ICIs and/or MTAs for advanced HCC. Due to the study design and the evaluation of HCC treatments, the number of patients included in this study is limited, especially in the SMI decrease group with a small sample. We also believe that this observation period is reasonable, since the patients in this study were advanced HCC patients with extensive intrahepatic involvement and/or with major vcascular invasion, and the expected survival time for such patients has been reported to be 3-5 months (J Hepatol. 57(4), 821-829, 2012).
As you pointed out, we have added this point as a limitation in the discussion section. Despite this limitation, we believe that the decrease SMI immediately after HAIC is a strong prognostic factor in HCC patients who underwent HAIC, and this study can provide novel evidence with a strong clinical impact.
- This retrospective cohort study recruited patients between January 2009 and January 2022. How has COVID-19 affected patient recruitment during the last three years of this period? This issue should be explained and discussed.
Response: Thank you very much for your comments.
Recent advances in systemic therapy for patients with advanced HCC have been remarkable, and based on the results of phase III trials, six regimens of sorafenib (Lancet Oncol. 10(1), 25-34, 2009), regorafenib (Lancet. 389(10064), 56-66, 2017), lenvatinib (Lancet. 391(10126), 1163-1173, 2018), cabozantinib (N Engl J Med. 379(1), 54-63, 2018), ramucirumab (Lancet Oncol. 20(2), 282-296, 2019), and atezolizumab plus bevacizumab (N Engl J Med. 382(20), 1894-1905, 2020) are currently approved for unresectable HCC. In addition, in the first regimen that does not include MTAs, tremelimumab plus durvalumab combination immunotherapy, has shown safety and efficacy in uHCC in phase III trials (NEJM Evid. 1(8), 2022). With this expanded clinical use of systemic therapies, the opportunities to perform HAIC or transcatheter arterial chemoembolization (TACE). In fact, in the last five years since 2017, when the number of available systemic therapies began to increase, the number of eligible patients for this study was halve.
On the other hand, the global pandemic spread of COVID-19 over the past three years has not resulted in a dramatic decrease in the number of patients with HCC undergoing HAIC, and the number of HCC patients treated with other therapies including radiofrequency ablation and TACE has also not changed. Our institution was able to continue treating HCC patients without being affected by the COVID-19 epidemic. Therefore, we believe that the recent decrease in the number of patients treated with HAIC is mainly due to the spread of systemic therapy other than HAIC for HCC, rather than the COVID-19 epidemic. Although these trends in HCC treatments have changed over time, HAIC remains a promising treatment for advanced intrahepatic lesions, especially with major vascular invasion, which are difficult to treat with general systemic therapies, and this study can be novel evidence of a strong clinical impact.
We agree with you that patient recruitment in this study may have been influenced by the time, so we have added this point as a limitation in the discussion section.
- In the Methods-2.3. Evaluation of Parameters, the authors described “Muscle mass was assessed by measuring the skeletal muscle and psoas muscle area at the level of the third lumbar vertebra and dividing these areas by the square of the height, SMI, and psoas muscle index (PMI)’. The so-called “skeletal muscle” here refers to what parts are measured? Does it contain psoas muscle? Why is the psoas muscle part measured in addition? These issues should be clarified.
Response: Thank you very much for your comments.
Skeletal muscle area at the third lumber level are collective term that includes the psoas muscle, the iliopsoas muscle, and the muscles that make up the abdominal wall. Muscle mass was assessed by measuring the skeletal muscle and psoas muscle area at the level of the third lumbar vertebra and dividing these areas by the square of the height to calculate skeletal muscle index (SMI) and psoas muscle index (PMI). As you pointed out, we have revised the corresponding part of the method section to clarify the meaning.
PMI is useful for the assessment of muscle mass, and it has been reported that PMI correlates with SMI measured by bioelectrical impedance analysis (BIA) and CT findings (Minerva Gastroenterol Dietol. 59(2), 173-186, 2013). PMI is versatile in actual clinical practice because it can be measured simply without the use of dedicated muscle mass measurement software.
As shown in Figure 2 of this paper, PMI was positively correlated with SMI in advanced HCC patients who underwent HAIC in our study as well.
We agree with your opinion that PMI and SMI are similar muscle measures and strong confounders, so we have reanalyzed the multivariate analysis on PFS and OS excluding PMI as an explanation variable. The results of the reanalysis identified decreased SMI at 3 weeks after HAIC as a significant independent factor associated with shorter PFS and OS.
To reflect these changes, we have revised the corresponding part of the results section, Table 4, and Table 5.
- In Table 3, the results showed that the number of samples was too small to make the AEs results unreliable. The authors should give a reasonable explanation.
Response: Thank you very much for your comments.
While general molecular targeted therapies are intended for advanced HCC patients with preserved liver function in Child-Pugh A, HAIC is a safe treatment that is well tolerated in HCC patients with impaired liver function, such as Child-Pugh B. As Table 3 shows, HAIC is associated with a wide variety of AEs, each of which is less frequent and less severe than that of molecular targeted therapies. In comparing the frequency of AEs in the decreased SMI and non-decreased SMI groups, we calculated significant differences using Fisher’s exact test, a test method used in the analysis of two categories of data when the sample size is small (Statistical Science. 7(1), 131-153, 1992).
HAIC is a treatment technique with many differences depending on the institution where it is performed and the regimen used. Although future studies with large sample sizes including multicenter cases are need, this study was designed to examine a limited sample size at a single institution in a retrospective manner to eliminate technical bias.
We agree with you that small sample size makes it difficult to compare the occurrence of AEs in HAIC in the decreased SMI and non-decreased SMI groups. Therefore, we have added this point as a limitation in the discussion section.
- In Figure 4, the meanings for star symbols (* and **) should be clearly described.
Response: Thank you very much for your comments.
In Figure 4, *P<0.05 and **P<0.01 are shown.
To clarify the meaning, we have added the corresponding part to the Figure legend to clarify the meaning.
- In Tables 4 and 5, what are the recruited patient numbers?
Response: Thank you very much for your comments.
Both Tables 4 and 5 show the results of the multivariate analysis for all 70 patients who enrolled in this study. We have added the number of patients to both tables to present the results correctly.
- In Abstract, line 26, “skeletal muscle mass (SMI)” can be revised.
Response: Thank you very much for your comments.
We have revised the corresponding part of abstract.

Round 2
Reviewer 1 Report
The authors answered our comments sincerely. I consider that this manuscript is suitable for publication
Reviewer 2 Report
The revised version of the paper is OK. Thank you!
Reviewer 3 Report
No further comments. This revised manuscript can be accepted.